

# Soil moisture–induced changes in land carbon sink projections in CMIP6

Lea Maria Gabele[1], Petra Sieber[1], Laibao Liu[2], and Sonia Isabelle Seneviratne[1]

[1]Institute for Atmospheric and Climate Science, Department of Environmental Systems Science, ETH Zurich, Zurich, Switzerland
[2]Department of Geography, University of Hong Kong, Hong Kong SAR, China

**Correspondence:** Lea Maria Gabele (lea.gabele@usys.ethz.ch) and Petra Sieber (petra.sieber@env.ethz.ch)

**Abstract.** The terrestrial biosphere absorbs about one third of anthropogenic $CO_2$ emissions, thereby significantly slowing human–induced climate change. Its capacity to act as a carbon sink strongly depends on climate conditions, particularly soil moisture (SM), which can constrain plant growth and amplify land–atmosphere feedbacks. Therefore, accurately capturing these effects in Earth System Models (ESMs) is critical.

Using dedicated experiments of the Land Feedback Intercomparison Project (LFMIP, an experiment within the Land Surface, Snow, and Soil Moisture Model Intercomparison Project, LS3MIP) from the latest generation of ESMs from the Coupled Model Intercomparison Project Phase 6 (CMIP6), we show that projected SM changes substantially reduce the land carbon sink by the end of the century (2070–2099). This reduction is mainly driven by SM variability, highlighting the importance of SM extremes, which are projected to become more frequent and intense under climate change. Our results confirm those of

the previous model generation based on the Global Land-Atmosphere Climate Experiment–Coupled Model Intercomparison Project phase 5 (GLACE–CMIP5). The results show that the strong negative impact of SM changes on the land carbon sink shown for GLACE–CMIP5 is less severe in LFMIP. A more in–depth analysis reveals that this is due at least in part to the specific ESM sampling of the respective experiments, with participating ESMs from CMIP5 generally showing a stronger drying trend. Despite agreement on the negative impact of SM on the land carbon sink in most tropical and mid–latitude

ecosystems in both sets of multi–model experiments, there are large intermodel differences in the projected magnitudes.

As SM can influence land carbon uptake both directly and indirectly via land–atmosphere coupling, we conduct a contribution analysis on the impact of direct and indirect SM effects on carbon uptake, which reveals that SM–atmosphere interaction dominate SM–induced changes globally. However, models show disagreement on the magnitude of these effects. Intermodel differences arise mainly from varying sensitivities of GPP to SM–related direct and indirect effects, suggesting that differences

likely stem from varying representations of water–stress related processes across ESMs.

Our findings highlight SM–atmosphere coupling as a critical factor for future land carbon uptake. Improving the representation of water stress processes, plant hydraulics, and vegetation characteristics in ESMs is essential for reducing uncertainty in projections. Maintaining and possibly extending the experimental set up to a larger set of models in future CMIP generations will be key to advancing our understanding of SM–carbon interactions and consequently of the evolution of the land carbon

sink under human–induced climate change.





## 1 Introduction

The terrestrial biosphere plays a crucial role in Earth's climate system by acting as a net carbon sink. The land carbon sink is determined by the balance of carbon uptake through photosynthesis (Gross Primary Production, GPP) and losses via ecosystem respiration and disturbances such as wildfires and land–use change (Keenan and Williams, 2018). The net carbon exchange on
large spatial and temporal scales is referred to as Net Biome Production (NBP). In the recent past, the global land carbon sink has increased with increasing atmospheric $CO_2$ concentration and has absorbed about one third of anthropogenic $CO_2$ emissions (Friedlingstein et al., 2025). This underscores the important role of the land carbon sink in dampening the atmospheric $CO_2$ growth rate, thus slowing down human–induced climate change (Friedlingstein et al., 2006).

The capacity of terrestrial ecosystems to remove and store carbon is strongly influenced by climate conditions. Tempera-
ture–induced increases in respiration, thawing permafrost, decreasing water availability, and the impact of more intense and frequent extreme events on ecosystems weaken the carbon sink and amplify climate change (Arora et al., 2020; Natali et al., 2021; Reichstein et al., 2013; Williams et al., 2019). Evidence on the importance of water–carbon coupling in modulating the evolution of the land carbon sink has grown (Gentine et al., 2019; Green et al., 2019; Huang et al., 2016; Humphrey et al., 2018; Liu et al., 2020, 2023; Stocker et al., 2018; Sun et al., 2025). The effect of water availability on NBP is particularly
evident in regions where land carbon uptake is restricted by moisture availability (water limitation) rather than by net radiation (energy limitation) (Green et al., 2019; Marcolla et al., 2020; Seneviratne et al., 2010).

Recent studies have shown that decreased soil moisture (SM) availability plays a crucial role in weakening the land carbon sink. It has been identified as the dominant cause of interannual carbon flux anomalies (Humphrey et al., 2021; Liu et al., 2024), but also affects long–term changes in the land carbon sink because the negative effect of water stress on ecosystems is not offset
by subsequent equivalent wet anomalies (Green et al., 2019). As the frequency and intensity of extreme events is projected to increase with climate change (IPCC, 2023), this could substantially weaken the land carbons sink in the future. Prolonged (SM) droughts weaken ecosystem resilience, raising susceptibility to pathogens, insect outbreaks, and plant mortality, and increasing the risk of fire events (Allen et al., 2015; Seidl et al., 2017; Sippel et al., 2018; Williams and Abatzoglou, 2016). This impairs drought recovery and can lead to lower carbon storage capacity (Green et al., 2019; Reichstein et al., 2013; Zheng
et al., 2021). Consequently, water scarcity can have immediate, delayed, and potentially long–lasting effects on the land carbon sink (Anderegg et al., 2015; Kannenberg et al., 2020; Schwalm et al., 2017).

Decreased SM can cause water stress in vegetation due to direct physiological effects of water limitation on GPP and indirect effects via land–atmosphere feedbacks (Green et al., 2019; Humphrey et al., 2021; Liu et al., 2025; Zhou et al., 2019). Direct SM effects arise from photosynthetic activity being constrained by soil water availability. Under dry conditions, plants partially
close their stomata to limit water losses through transpiration, which in turn reduces $CO_2$ uptake and thus limits GPP (Sippel et al., 2018). Indirect SM effects arise from interactions between the land surface and the atmosphere. Low SM suppresses latent heat flux and evapotranspiration, leading to higher surface temperatures and vapour pressure deficit (VPD). By further reducing SM, these effects create a feedback loop that intensifies water stress in plants and exacerbates the direct SM effect (Berg et al., 2016; Seneviratne et al., 2010).





To assess the representation of SM–atmosphere coupling and its long–term effect in climate projections, the Global Land–Atmosphere Coupling Experiment of the Coupled Model Intercomparison Project Phase 5 (GLACE–CMIP5) was introduced (Seneviratne et al., 2013). Using the GLACE–CMIP5 experiment, Green et al. (2019) demonstrated that SM changes have a significant long–term impact on NBP under a high–emission scenario, with changes in SM reducing the carbon sink by half its potential magnitude by the end of the 21st century (Green et al., 2019). However, SM–induced changes in the land carbon sink show large intermodel differences and the origins of these differences remain elusive.

In this study, we assess the impact of SM on the land carbon sink in the latest generation of Earth System Models (ESMs) from the Coupled Model Intercomparison Project Phase 6 (CMIP6) using dedicated experiments of the Land Feedback Intercomparison Project (LFMIP) and compare it with results from GLACE–CMIP5. Furthermore, we conduct a contribution analysis to quantify the extent to which SM–induced changes in land–carbon dynamics emerge through direct linear effects of water limitation on photosynthesis, versus indirect effects mediated by land–atmosphere feedbacks and investigate the origins of intermodel differences. Thereby, our analysis aims to further improve the understanding of SM–carbon coupling in state–of–the–art ESMs and its implications for long–term carbon cycle projections. Gaining a deeper understanding of the moderating processes within the land carbon cycle is essential for enhancing projections of atmospheric $CO_2$ growth rates, which in turn is crucial for accurately estimating the magnitude of future climate change (Arora et al., 2020; Friedlingstein et al., 2025).

## 2 Data and methods

### 2.1 Model experiments and data

To assess SM–induced changes in the land carbon sink, ESM output data of LFMIP are used, which is part of the Land Surface, Snow, and Soil Moisture Model Intercomparison Project (LS3MIP, Van Den Hurk et al., 2016) under CMIP6. LFMIP follows the GLACE–CMIP5 blueprint (Seneviratne et al., 2013) and is designed to diagnose changes in land–atmosphere coupling related to SM.

LFMIP consists of three experiments, a reference run (CTL) based on the historical and the Shared Socioeconomic Pathway 5–8.5 (SSP5–8.5) scenarios of CMIP6 and two experiments where SM is prescribed as (i) the mean seasonal cycle of 1980–2014 (pdLC) and (ii) the 30–year running mean (rmLC) of CTL (see supplementary Figure S1 for an illustration of the experiments). LFMIP outputs are available from the four ESMs of CMIP6, CESM2, IPSL–CM6A–LR, MPI–ESM1–2–LR, and CMCC–ESM2 (for detailed information see supplementary Table S1). LFMIP covers the period 1981–2099 at monthly resolution.

The SM–induced changes in NBP derived from LFMIP are compared with those of the previous generation, GLACE–CMIP5. For this purpose, we partially reproduce and build on the analysis of Green et al. (2019) to compare SM–induced changes in NBP between model generations for the period 1981–2099. Available for the analysis of GLACE–CMIP5 are outputs from four models, CESM, GFDL, IPSL, and MPI–ESM (see Seneviratne et al. (2013) Table 1 for detailed information). Among the available ESMs for each model generation, three of four model are from the same modelling group. The GLACE–CMIP5



experiment follows the same protocol as LFMIP for the previous model generation, CMIP5. To compare recent and projected future conditions across models, our analysis focuses on a baseline period (1981–2010) and a end–of–century future period (2070–2099). We note that the period for the SM prescription in GLACE–CMIP5 (1971–2000) is roughly one decade earlier than in LFMIP. This impedes the direct comparison for the impact of SM on NBP for the baseline period, but our focus lies on the evolution of SM–induced impacts on NBP by the end of the century, which are not affected. For spatially resolved comparisons across models, the data is regridded to 1° resolution using bilinear interpolation.

Larger sets of CMIP6 (Eyring et al., 2016) and CMIP5 (Taylor et al., 2012) models are used to assess if the available models of LFMIP and GLACE–CMIP5 are representative of the respective CMIP generation. Ten ESMs of CMIP5 (i.e., BNU–ESM, CESM1–BGC, CanESM2, CanESM2, GFDL–ESM2G, HadGEM2–CC, HadGEM2–CC, IPSL–CM5A–LR, MPI–ESM–LR, and NorESM1–M) and nine ESMs of CMIP6 (i.e., ACCESS–ESM1–5, CESM2, CMCC–ESM2, EC–Earth3–Veg–LR, GFDL–ESM4, IPSL–CM6A–LR, MPI–ESM1–2–LR, ’MRI–ESM2–0, NorESM2–LM) were used for this comparison, selecting one ensemble member from each modelling group.

We use ESM output for NBP to assess the land carbon sink, GPP to assess land carbon uptake, and total column soil moisture to quantify changes in projected SM. Since ESMs account for different numbers of soil layers and also vary in soil layer depth, output data for SM varies substantially in magnitude. Therefore, we standardise SM output to z–score values, using

$$z = \frac{X - \mu_{ref}}{\sigma_{ref}} \tag{1}$$

where each data point $X$ is standardised by the mean of the reference period $\mu_{ref}$ and its standard deviation $\sigma_{ref}$. As reference period we use the baseline period (1981–2010) to emphasize changes in SM compared to the baseline period from which SM is prescribed in the experiments.

For the contribution analysis of direct and indirect SM effects (see Sect. 2.3) we further use LFMIP model output at monthly resolution for the variables near–surface air temperature (T, in °C), surface downwelling short–wave radiation (R, in $Wm^{-2}$) and vapour pressure deficit (VPD, in kPa). VPD is calculated as the difference between the saturated water vapour pressure ($e_s$, in kPa) and the actual water vapour pressure ($e_a$, in kPa), where $e_s$ is determined by T using the Clausius–Clapeyron relation, such that

$$e_s = 0.6108 \times \exp\left(\frac{17.27 T}{T + 237.3}\right)$$

and $e_a$ is calculated from relative humidity (RH, in %) as

$$e_a = \frac{\text{RH}}{100} \times e_s.$$

## 2.2 Isolating the effects of soil moisture

To isolate the effects of SM on NBP (and similarly, on GPP), a method commonly used in experiments following the GLACE blueprint is employed (see (Green et al., 2019; Seneviratne et al., 2013). Changes in NBP ($\Delta NBP_{CTL}$) can be described as

$$\Delta NBP_{CTL} = \Delta NBP_{SMtrend} + \Delta NBP_{SMvar} + \Delta NBP_{other} \tag{2}$$





where $\Delta NBP_{SMtrend}$ is the effect of changes in mean SM conditions on NBP and $\Delta NBP_{SMvar}$ the effect of SM variability on NBP. $\Delta NBP_{other}$ summarises changes in NBP due to the fertilisation effect of $CO_2$, temperature changes, and other influencing factors. The experiments of LFMIP allow isolating the effect of SM trend and variability, where $\Delta NBP_{SMtrend} = \Delta NBP_{rmLC-pdLC}$ and $\Delta NBP_{SMvar} = \Delta NBP_{CTL-rmLC}$, as well as the combined effects of SM expressed as $\Delta NBP_{SMall} = \Delta NBP_{SMtrend} + \Delta NBP_{SMvar} = \Delta NBP_{CTL-pdLC}$.

## 2.3 Separating the contributions of direct and indirect soil moisture effects

To assess the impact of direct and indirect SM effects, we focus on GPP which is more strongly influenced by SM changes than respiration, particularly in regions that strongly contribute to global NBP (see supplementary Fig. S2 and S5). SM–induced changes in GPP ($\Delta GPP$) can occur as direct effect through changes in water availability for photosynthesis and as indirect SM effect via SM–atmosphere coupling. To assess the contribution of direct and indirect effects, we conduct a contribution analysis. Following the approach of Humphrey et al. (2021), a multiple linear regression is performed to capture the local direct and indirect effects of SM on GPP for monthly time steps on grid cell level, using

$$\Delta GPP^* = \beta_{SM}\Delta SM + \beta_T\Delta T + \beta_{VPD}\Delta VPD + \beta_R\Delta R, \tag{3}$$

where $\Delta GPP^*$ describes the estimated SM–induced change in GPP due to $\Delta SM$, which represents the change in SM; $\Delta T$ denotes the SM–induced change in near–surface air temperature, $\Delta VPD$ that in vapour pressure deficit, and $\Delta R$ that in downward solar radiation, respectively, and $\beta$ are the corresponding regression coefficients. The regression is performed over 30 years and the resulting impacts of the individual drivers SM, T, VPD, and R are referred to as $\Delta GPP^*_{SM}$, $\Delta GPP^*_T$, $\Delta GPP^*_{VPD}$, and $\Delta GPP^*_R$, respectively. Following the reasoning of Humphrey et al. (2021), the impact of indirect SM effects via T and VPD ($\Delta GPP^*_T$ and $\Delta GPP^*_{VPD}$) are reported as their sum ($\Delta GPP^*_{T\&VPD}$), since the calculation of VPD depends on T, limiting the ability to disentangle their impact on GPP. Their individual effects are reported in supplementary Fig. S12 and S13 for completeness.

We note that here we focus only on the total SM effect (and not on the individual effects of SM trend and variability), thus, $\Delta GPP$ is equivalent to $GPP_{CTL} - GPP_{pdLC}$ and consequently the regression estimates also refer to the total SM effect on GPP. For better readability we omitted the subscript "SMall" for this part of the analysis.

## 2.4 Measures of intermodel differences

To assess whether differences in projected direct and indirect SM effects or in the sensitivity of GPP to those SM effects dominate intermodel spread, we perform a factorial analysis of variance (ANOVA) on the SM effects derived by regression, i.e., $\Delta GPP^*_{SM}$, $\Delta GPP^*_T$, and $\Delta GPP^*_{VPD}$ (see Sect. 2.3). We neglect $\Delta GPP^*_R$ due to its small impact on global $\Delta GPP^*$ (see Sect. 3.2.1). We constructed a full factorial ensemble of $\Delta GPP^*$ by systematically combining the projected SM effects and sensitivities of GPP across LFMIP models. At each grid cell, the SM–induced change in GPP for a given combination of factor levels is expressed as

$$\Delta GPP^*_{ijklmn} = \beta_{SM_i}\Delta SM_j + \beta_{T_k}\Delta T_l + \beta_{VPD_m}\Delta VPD_n + \varepsilon_{ijklmn}, \tag{4}$$



where $\Delta SM$, $\Delta T$, and $\Delta VPD$ denote the direct and indirect SM effects and $\beta_{SM}$, $\beta_T$, and $\beta_{VPD}$ are the sensitivities of GPP to the respective SM effect. The indices $i, j, k, l, m, n$ denote the levels for each of the six factors (i.e., the individual ESMs) from which the SM effects and sensitivities are drawn. The factorial ANOVA then attributes the variance in $\Delta GPP^*$ across factor combinations to (i) the sensitivity of GPP to SM, T, and VPD and (ii) the projected changes in SM, T, and VPD. This decomposition allows us to quantify the relative contributions of differences in projected SM effects and in the associated sensitivities of GPP to the intermodel spread of SM–induced impacts on GPP.

## 3 Results

### 3.1 Comparison of SM–induced impacts in GLACE–CMIP5 and LFMIP

We assess the impact of SM on NBP for the latest CMIP generation using the available LFMIP models, and compare the results with those from GLACE–CMIP5 (Fig. 1). The LFMIP model mean projects the global land carbon sink (global $NBP_{CTL}$) to increase from $1.07 \pm 0.11$ PgC yr$^{-1}$ (model mean $\pm$ standard deviation across models) during the baseline period (1981–2010) to $2.91 \pm 1.38$ PgC yr$^{-1}$ by the end of the century (2070–2099, Fig. 1a and c). The GLACE–CMIP5 mean for global $NBP_{CTL}$ is of similar magnitude as that of LFMIP for the baseline period ($1.23 \pm 1.38$ PgC yr$^{-1}$). However, GLACE–CMIP5 projects a decline in global $NBP_{CTL}$ after 2070, resulting in only a small increase in the land carbon sink by the end of the century ($1.51 \pm 1.69$ PgC yr$^{-1}$ (Fig. 1b and c)), whereas LFMIP projects the global $NBP_{CTL}$ to almost triple. Thus, the LFMIP mean projects a stronger land carbon sink by the end of the century compared to the GLACE–CMIP5 mean. However, the significantly smaller global $NBP_{CTL}$ in GLACE–CMIP5 is influenced by the CESM model, which projects NBP to be a consistent net carbon source over the 21st century (supplementary Fig. S3a).

In both model generations, projected SM changes negatively affect global $NBP_{CTL}$ (Fig. 1). For the LFMIP mean, the projected negative effect of SM on NBP is overall less severe compared to GLACE–CMIP5 (Fig. 1a and b). GLACE–CMIP5 models project the global land carbon sink to be reduced by about twice its absolute magnitude during the baseline period ($-2.44 \pm 1.77$ PgC yr$^{-1}$), as previously shown by (Green et al., 2019) (Fig. 1b). This is primarily due to the negative impact of SM variability ($-2.21 \pm 1.67$ PgC yr$^{-1}$). While the impact of the SM trend leads to a growing reduction over the 21st century (to $-1.14 \pm 1.09$ PgC yr$^{-1}$ by the end of the century), the negative effect of SM variability is projected to decline (to $-1.26 \pm 0.41$ PgC yr$^{-1}$ by the end of the century, Fig. 1c). The total impact of SM on NBP at the end of the century reduces the global land carbon sink to half. In LFMIP, the projected negative effect of SM on NBP is overall less severe (Fig. 1a). In the baseline period, SM changes reduce the global land carbon sink by $-0.69 \pm 0.15$ PgC yr$^{-1}$. In the future period, the negative impact of SM changes increases in absolute magnitude to $-1.22 \pm 1.04$ PgC yr$^{-1}$ (Fig. 1c). The effect of SM variability remains relatively unchanged throughout the century, reducing NBP by $-0.66 \pm 0.22$ PgC yr$^{-1}$ in the baseline period and $-0.82 \pm 0.80$ PgC yr$^{-1}$ in the future period. The impact of SM trend becomes increasingly negative, resulting in a reduction of $-0.40 \pm 0.27$ PgC yr$^{-1}$ by the end of the century. Nevertheless, in relative terms the negative impact of SM reduces global $NBP_{CTL}$ by 64 % for the baseline period and only by 42 % during the future period, because global $NBP_{CTL}$ of LFMIP is projected to increase over time.



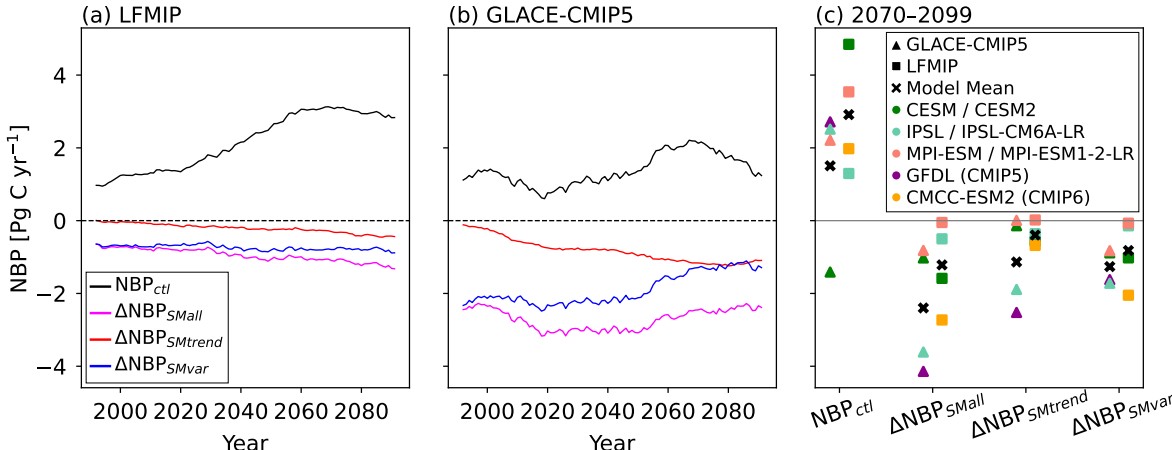

**Figure 1.** Comparison of the global evolution of NBP in (a) GLACE–CMIP5 and (b) LFMIP from 1991 to 2089 (smoothed with a 20 year centred rolling mean), including total NBP (NBP$_{CTL}$, black), as well as the changes in NBP due to SM trend (ΔNBP$_{SMtrend}$, red), SM variability (ΔNBP$_{SMvar}$, blue) and the total SM effect (ΔNBP$_{SMall}$, pink), and (c) individual model projections of global NBP an the respective SM–induced changes for the future period (2070–2099) for GLACE–CMIP5 (triangles) and LFMIP (squares). For panels (a) and (b) the colour scheme was adapted from Green et al. (2019) to facilitate comparison with their study.

Assessing spatial patterns of the effect of SM on NBP shows a reduction of NBP due to SM in most areas for both model generations, mainly due to the effect of SM variability (Fig. 2, supplementary Fig. S6). The reduction is strongest in tropical regions of South America and in the mid–latitudes of the northern hemisphere, where NBP$_{CTL}$ is generally high (Fig. 4, supplementary Fig. S5). This SM–induced reduction in large parts of the globe is already apparent during the baseline period due to the negative impact of SM variability (supplementary Fig. S8 and S9). While LFMIP projects the negative effect of SM on NBP to intensify further over the century, especially in tropical regions and mid–latitudes of the northern hemisphere (Fig. 1a, supplementary Fig. S9), this is not the case in GLACE–CMIP5, where several regions show a substantial reduction in the negative effect of SM, especially in the mid and high latitudes of the northern hemisphere (Fig. 1b, supplementary Fig. S8). However, in both CMIP generations, models show high disagreement on the sign of change of the SM–induced impact on NBP for the future period relative to the baseline period in several regions, including regions where models agree on the projected change in SM (Fig. 3). For both CMIP generations, intermodel differences in SM–induced changes in NBP are mainly located in tropical regions and the northern mid–latitudes.

Comparing the SM evolution in GLACE–CMIP5 and LFMIP reveals substantial differences in the magnitude and spatial extent of the projected SM drying (Fig. 3). The GLACE–CMIP5 mean shows widespread SM drying across the globe (Fig. 3a.1), which is less pronounced in LFMIP (Fig. 3b.1). The GLACE–CMIP5 models IPSL and GFDL (Fig. 3a.3 and a.5) project severe SM drying in several areas, especially in the northern mid–latitudes by the end of the century, resulting in a strong negative impact of SM on NBP. The CMIP6 version of IPSL included in LFMIP, IPSL–CM6A–LR (Fig. 3b.3), shows





**Figure 2.** Changes in NBP due to the total SM effect ($\Delta NBP_{SMall}$) for the future period (2070–2099) for (a.1) the GLACE–CMIP5 mean, (b.1) the LFMIP mean, (a.2–5) individual model projections of GLACE–CMIP5, and (b.2–5) individual model projections of LFMIP. Pink indicates a reduction and green an increase of NBP due to SM. White indicates no data.



**Figure 3.** SM from the control simulation for the future period (2070–2099) for (a.1) the GLACE–CMIP5 mean, (b.1) the LFMIP mean, (a.2–5) individual model projections of GLACE–CMIP5, and (b.2–5) individual model projections of LFMIP. Brown indicates drying and green wetting relative to the pre–industrial period. White indicates no data.



less SM drying in most areas than IPSL and even a reversed trend (i.e., SM wetting) in several regions, including vast areas of Central and Southeast Asia, Northern Europe, Central Africa, and North America. Similarly, MPI–ESM1–2–LR (LFMIP, Fig. 3b.4) shows less pronounced SM drying than MPI–ESM (GLACE–CMIP5, Fig. 3a.4) and a reversed SM trend in large parts of Central and Southern Africa, as well as in Western and Southeast Asia, leading to a less negative SM–induced impact on

NBP. CESM2 (LFMIP, Fig. 3b.2) and CESM (GLACE–CMIP5, Fig. 3a.2) generally show less pronounced differences in their SM projections.

We assess the representativeness of the available models in GLACE–CMIP5 and LFMIP for the respective CMIP generation by comparing the multi–model means (MMMs) for latitude zones for NBP and SM (Fig. 4). LFMIP and GLACE–CMIP5 have a similar mean and spread (95 % confidence interval) as the respective full ensemble in most latitude zones. However,

GLACE–CMIP5 shows stronger SM drying in the mid–latitudes of the northern hemisphere than the CMIP5 MMM, primarily due to GFDL and IPSL (Fig. 4b.1, supplementary Fig. S10 b.2). As the northern mid–latitudes are the main contributors to global NBP (accounting for 41 % (45 %) of global NBP in LFMIP (CMIP6 MMM) and 70 % (85 %) in GLACE–CMIP5 (CMIP5 MMM)), this must be considered when interpreting the strong negative impact of SM on NBP projected by these models in GLACE–CMIP5.

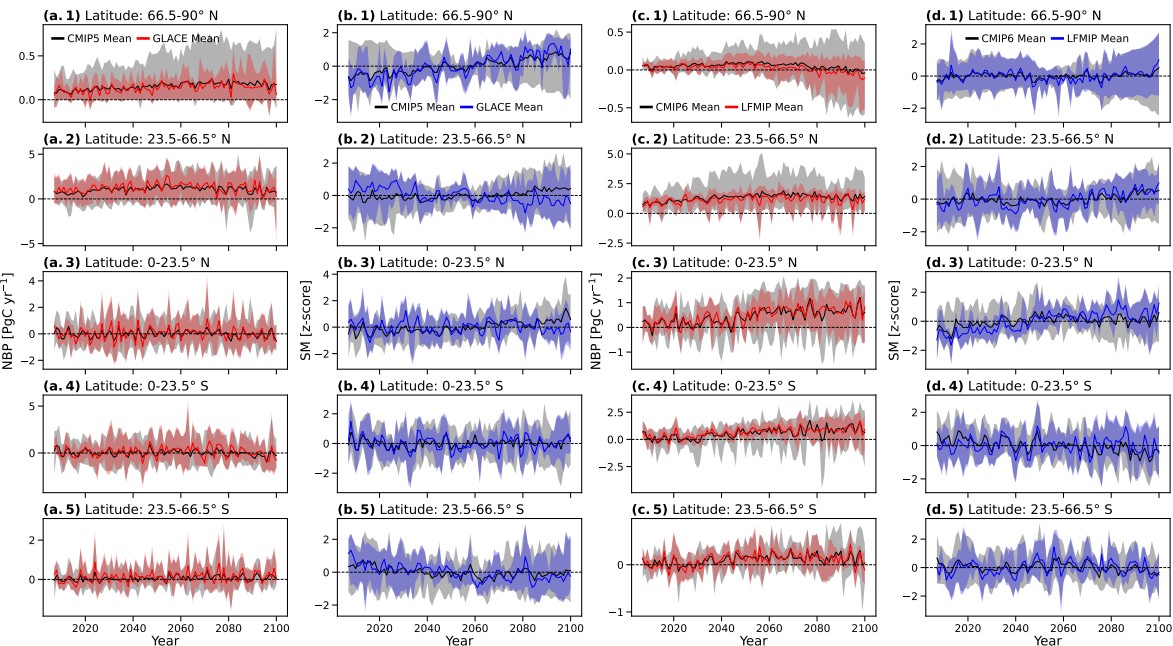

**Figure 4.** Comparison of NBP and SM of (a, b) GLACE–CMIP5 to the CMIP5 multi–model mean (MMM) and (c, d) LFMIP to the CMIP6 MMM across latitude zones of 30° from north (1) to south (5). In columns (a) and (c), red lines show latitudinal NBP and red shading the 95 % confidence interval for GLACE–CMIP5 and LFMIP, respectively. In Columns (c) and (d), blue lines show latitudinal SM and blue shadings the 95 % confidence interval. Black lines indicate the full ensemble MMM and grey shadings the 95 % confidence interval.





## 3.2 Origins of SM–induced changes in land carbon uptake


With focus on LFMIP, we further analyse the origins of SM–induced changes in land carbon uptake by conducting a contribution analysis to assess the impact of direct and indirect SM effects on GPP. Further, we investigate causes of intermodel differences in SM–induced changes in GPP from direct and indirect SM effects and their respective sensitivities (results for GLACE–CMIP5 are displayed in supplementary Figs. S15 to S17). To assess the impact of SM on land carbon uptake under

future climate change, we focus on the future period (2070–2099) when impacts of both SM variability and SM trend come into play.

### 3.2.1 Contribution of direct and indirect effects

To isolate the contributions of the direct SM effect and indirect effects via SM–atmosphere coupling, we adapt the approach from Humphrey et al. (2021) and perform a multiple linear regression of the local response of GPP to the total impact of SM

($\Delta GPP$) on the predictors $\Delta SM$ (as direct effect), $\Delta T$, $\Delta VPD$, and $\Delta R$ (as indirect SM effects) as described in section 2.3. The estimated $\Delta GPP^*$ shows high spatial agreement with the modelled $\Delta GPP$ (spearman r = 0.93), capturing 83 % of the global $\Delta GPP$ for the LFMIP mean for the future period, confirming the validity of the approach (Fig. 5, supplementary Fig. S11).

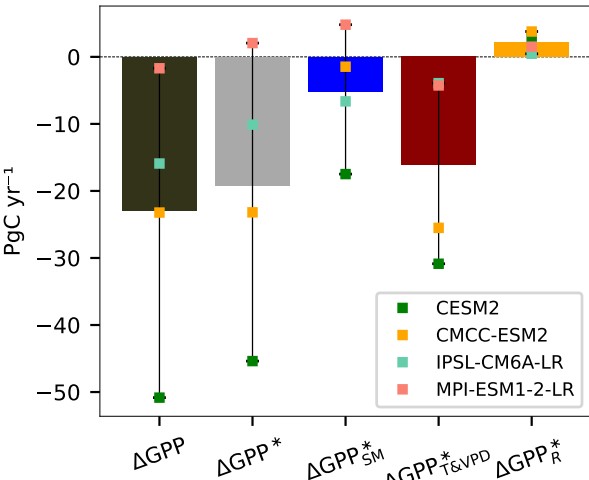

**Figure 5.** Global changes in GPP due to the total SM effect modelled in LFMIP ($\Delta GPP$, black bar). Contribution analysis estimates for LFMIP of the total global changes in GPP due to SM ($\Delta GPP^*$, grey bar) and individual contributions from the direct SM effect ($\Delta GPP^*_{SM}$, blue bar) and indirect SM effects via VPD and T ( $\Delta GPP^*_{T\&VPD}$, red bar) and R ($\Delta GPP^*_R$, yellow bar) for the future period (2070–2099). Bars show the LFMIP mean and coloured dots the individual models.

Globally, indirect effects via T and VPD ($\Delta GPP^*_{T\&VPD}$) dominate $\Delta GPP^*$, accounting for 84 % of the negative SM–induced

effect on the LFMIP mean, while the direct SM effect ($\Delta GPP^*_{SM}$) accounts for 27 % (Fig. 5) . In contrast, the indirect effect



on GPP via R ($\Delta GPP^*_R$) leads to a slight increase in GPP, but the impact is comparably small. For the LFMIP mean, the spatial results of the contribution analysis show that most areas that are projected to experience severe SM–induced reduction in GPP are dominated by $\Delta GPP^*_{T\&VPD}$ (Fig. 6). However, this is not reflected in all LFMIP models, as they disagree on whether direct or indirect effects dominate $\Delta GPP^*$.

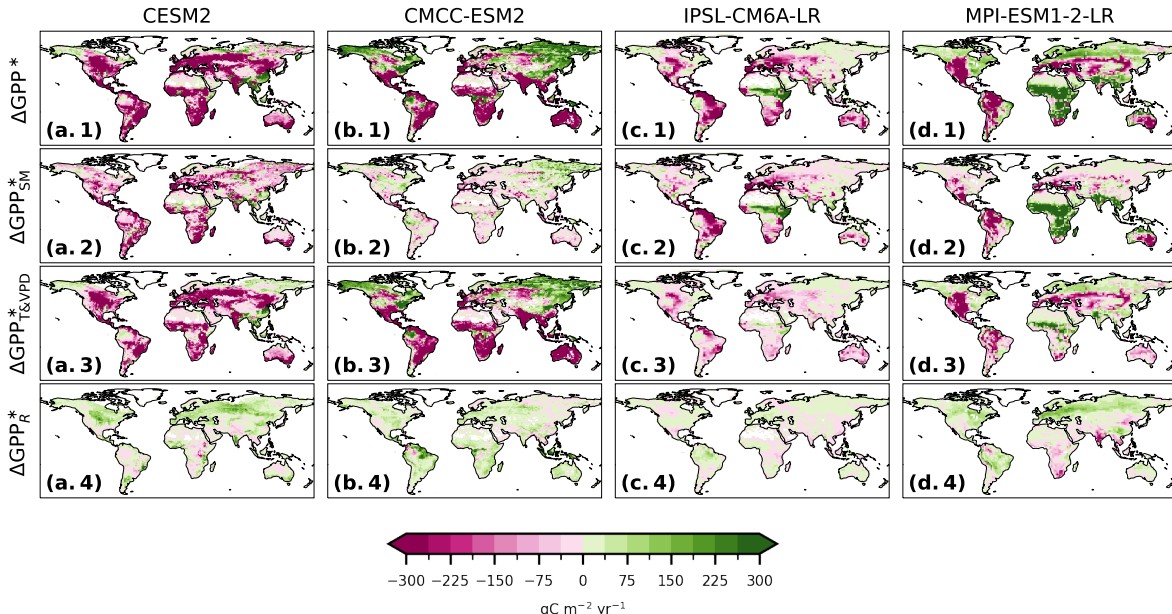

**Figure 6.** Contribution analysis estimate for LFMIP of the total changes in GPP due to SM ($\Delta GPP^*$) and the individual contributions from the direct SM effect ($\Delta GPP^*_{SM}$), the indirect SM effects via VPD and T ($\Delta GPP^*_{T\&VPD}$), and via R ($\Delta GPP^*_R$) for the future period (2070–2099). Pink indicates a reduction and green an increase of NBP due to SM. White indicates no data.

The results for CMCC–ESM2 and CESM2 exhibit substantially larger contributions of $\Delta GPP^*_{T\&VPD}$ (Fig. 6a and b), while IPSL–CM6A–LR and MPI–ESM1–2–LR show a larger relative contribution from the direct effect (Fig. 6c and d). For CMCC–ESM2, 109 % of the total negative contribution is attributed to $\Delta GPP^*_{T\&VPD}$ (offset by a slightly positive $\Delta GPP^*_R$, with the total indirect effects accounting for 93 % of the reduction in global GPP), making it the dominant impact in all regions affected by SM–induced changes in GPP. The results for CESM2 also suggest that $\Delta GPP^*_{T\&VPD}$ is the main

contribution to $\Delta GPP^*$ (accounting for 68 % of the reduction in GPP), with some regions in the tropics and mid–latitudes also showing a strong negative impact caused by the direct SM effect. For IPSL–CM6A–LR and MPI–ESM1–2–LR, the direct SM effect are generally of great importance, especially in tropical and lower mid–latitudinal regions that show SM drying. For IPSL–CM6A–LR that is primarily North South America, Central America, the Mediterranean, and South Africa. Similarly, MPI–ESM1–2–LR experiences a strong negative impact of the direct SM effect due to SM drying in North South America and





Central America, however, SM enhances GPP directly in large parts of Central Africa, where MPI–ESM1–2–LR is projected to experience strong SM wetting.

### 3.2.2 Contributions to intermodel differences

The results presented in Sect. 3.1 and 3.2.1 show substantial intermodel differences in LFMIP, revealing that even in regions where models agree on SM drying, they still disagree on the magnitude of the resulting impact on GPP (and consequently
NBP). This raises the question to what extent differences in SM–induced changes of land carbon uptake arise from differences in the sensitivity of carbon uptake to SM or deviations in SM projections itself.

We perform a factorial ANOVA to spatially assess contributions of direct and indirect SM effects, as well as of the sensitivity of GPP to those drivers as described in 2.4. In areas that experience a strong reduction in GPP due to SM, the results indicate that about 70–90 % of intermodel difference can be explained by either changes in the direct and indirect SM effects or
the sensitivity of GPP to those effects (supplementray Fig. S14). Overall, disagreement in the sensitivity of GPP to SM effects contributes most strongly to intermodel differences across large parts of the globe, especially in regions with large contributions to global $\Delta GPP$ like North South America, North America and Europe, where models largely agree on SM drying (Fig. 7a.3). While disagreement in the sensitivity to the direct SM effect dominates in the tropics (Fig. 7a.1), disagreement in the sensitivity to T and VPD dominates in mid and high latitudes (Fig. 7a.2). A similar but less pronounced spatial pattern can be seen for the
contributions of changes in the direct and indirect SM effects themselves (Fig. 7b).

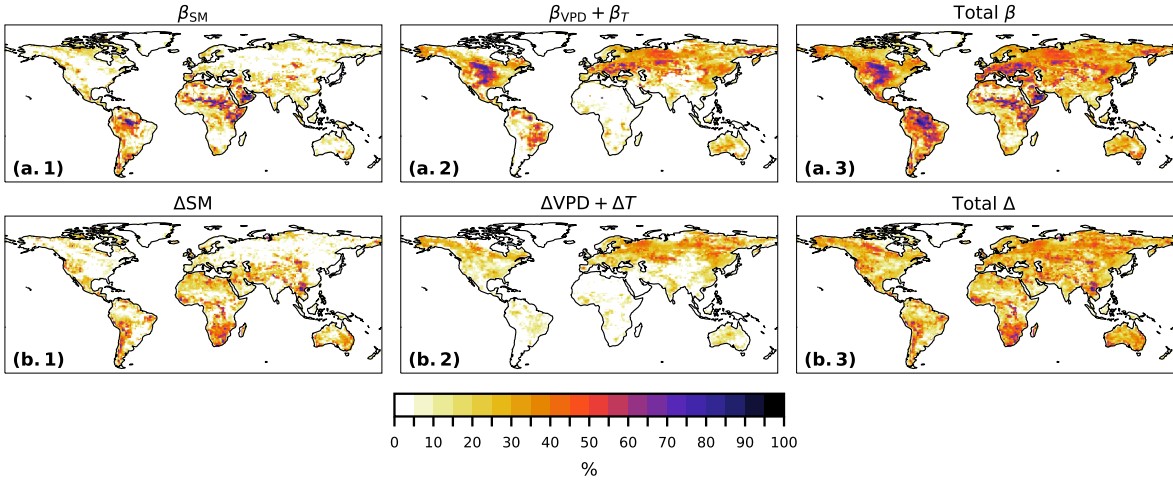

**Figure 7.** Contribution to intermodel differences (in %) in LFMIP from disagreement in the sensitivity of GPP to (a.1) the direct SM effect ($\beta_{SM}$), (a.2) indirect SM effects via T and VPD ($\beta_{VPD} + \beta_T$) and (a.3) direct and indirect SM effects (Total $\beta$), as well as contribution from (b.1) the change in SM ($\Delta SM$), (b.2) the change in T and VPD due to SM ($\Delta VPD + \Delta T$), and (b.3) the total contribution of change in SM (Total $\Delta$).





## 4 Discussion

### 4.1 Negative SM–induced impacts on the land carbon sink in LFMIP and GLACE–CMIP5

Our results show that projected SM changes negatively impact global NBP according to the LFMIP mean, which is consistent with previous findings based on GLACE–CMIP5 (Green et al., 2019). The negative effect is dominated by SM variability,

highlighting the strong negative impact of dry SM extremes on NBP. This finding aligns with observational evidence of the severe negative impact of extreme drought and heat events that lead to large carbon losses of terrestrial ecosystems (Ciais et al., 2005; Li et al., 2025; van der Woude et al., 2023). However, the negative SM effect on NBP is much stronger in the GLACE–CMIP5 mean, which is substantially influenced by two models, IPSL and GFDL, which also project particularly severe SM drying. Comparing SM projections of GLACE–CMIP5 to the full CMIP5 ensemble shows that this drying signal

is not representative of the model generation, as both models exhibit a strong drying in the mid–latitudes of the Northern Hemisphere not reflected in the CMIP5 MMM. The latitudinal NBP of the northern mid–latitudes accounts for about 85 % of global NBP for the CMIP5 MMM. Consequently, the stronger SM–induced reduction of NBP in GLACE–CMIP5 may be partly due to the model subset's bias toward stronger drying in these latitudes, suggesting that differences in the projected SM impact on NBP between GLACE–CMIP5 and LFMIP might thus be partly attributable to model selection rather than a

systematic differences between model generations.

Nevertheless, this does not imply that the overall conclusions of prior studies are overstated, because structural model limitations may lead to an overall underestimation of the severity of SM–induced reductions of NBP in ESMs of both generations. Previous studies highlighted that SM drought can strongly reduce carbon uptake, often followed by declining tree growth rates and increased tree mortality and potentially leading to lasting reductions in carbon uptake capacities of affected ecosystems

(Guo et al., 2025; Kannenberg et al., 2024; Kolus et al., 2019). This is not well captured by ESMs, because fundamental plant hydraulic properties are misrepresented, leading to shorter and weaker simulated drought impact on GPP than observed (Anderegg et al., 2015, 2020; Kolus et al., 2019). For example, a prominent study by Anderegg et al. (2015) showed that CMIP5 models do not capture the lagged effects of SM on NBP apparent in observational data. Among the models included in this study were also CESM and GFDL (which participated in the GLACE–CMIP5 experiment). Of all assessed CMIP5 models,

GFDL was shown to capture the lagged effects best. In addition to the projected strong SM drying, this further explains the strong negative impact of SM on NBP our study reveals for GFDL. In addition, declining moisture availability has also been linked to increased risk of fire (Byrne et al., 2024; Jones et al., 2024), but ESMs have been found to misrepresent the observed magnitude of carbon losses during fire events (Sanderson and Fisher, 2020). With compound fire weather and drought events increasing under climate change (Richardson et al., 2022), this may lead to carbon losses not captured by ESMs.

As the occurrence of negative SM extremes is projected to increase under climate change (IPCC, 2023), the inability of models to accurately capture these processes could lead to a growing underestimation of the severity of the negative SM–induced impacts on the future land carbon sink. Furthermore, Liu et al. (2023) demonstrate that the increased water–carbon coupling observed in the tropics (which implies increased carbon loss with declining water availability) is not well captured in state–of–the–art ESMs. Since ecosystems are projected to experience widespread shifts from energy to water limitation





throughout the 21st century (Denissen et al., 2022), this may further contribute to an underestimation of the simulated carbon loss under water stress.

## 4.2 SM–induced effects are dominated by atmospheric feedbacks

Isolating the contribution of direct and indirect SM effects demonstrates the importance of SM–atmosphere coupling in regulating ecosystem carbon fluxes, with the combined effect of T and VPD dominating the negative impact on global land
carbon uptake in LFMIP. However, models diverge substantially in the contribution of direct and indirect effects, with only MPI–ESM1–2–LR and IPSL–CM6A–LR showing higher contribution from direct SM effects, especially in tropical regions.

The debate on the main factor constraining land carbon uptake is ongoing, with conflicting results on whether the direct SM effect or effects of T and VPD via SM–atmosphere coupling dominate: On the one hand, direct SM limitation plays a key role in limiting carbon uptake (Kannenberg et al., 2024; Liu et al., 2020). This could imply that LFMIP models underestimate
the direct impact of soil water stress on photosynthesis, especially CESM2 and CMCC–ESM1. On the other hand, VPD has been identified as the dominant driver of drought–induced reductions in GPP, because increased VPD consistently limits photosynthesis, whereas SM only limits GPP below a certain threshold (Fu et al., 2022; Novick et al., 2016). Consequently, VPD exerts a more pervasive influence on GPP than the direct SM effect, which supports our assessment showing SM–atmosphere coupling as the key control on GPP globally.

Nevertheless, previous findings demonstrate that in moisture–limited regions the direct SM effect (i.e., SM availability limiting photosynthesis) becomes more important for constraining carbon uptake (Fu et al., 2022; Kannenberg et al., 2024; Liu et al., 2025). Our analysis demonstrates that LFMIP models fail to capture this consequence for regions that are known to be moisture–limited, which may partially be explained by how water stress is implemented in their land surface models. Stomatal conductance plays a central role in translating water limitation into reduced carbon uptake. In some land surface models the
stomatal conductance formulation is directly linked to VPD (Franks et al., 2018). Because of this formulation, the effect of increased T and VPD on stomatal behaviour may be projected to be more prominent in regulating GPP under SM drying, even if SM decline initiates the water stress response.

Among LFMIP models, CESM2 projects the strongest negative SM–induced impact on GPP. Its land surface model, CLM5 (the most recent version of the Community Land Model), offers the most advanced process implementations in terms of
realism by accounting for the complex coupling between SM, VPD and stomatal conductance. CLM5 includes new plant hydraulic processes such as leaf water potential, xylem water flow, and conductance loss, which are important for regulating the GPP response to dryness stress (Kennedy et al., 2019). The lack of explicit hydraulic constraints in most LFMIP models may limit their ability to capture the full impact of SM on GPP, indicating the need to further improve plant hydraulics in ESMs to accurately predict future GPP response to water stress, especially in regions with increased moisture limitation.
However, despite ongoing efforts, the complex mechanisms regulating ecosystems' response to drought are still not fully understood (Kannenberg et al., 2020), and the observed non–linear and species–specific responses of GPP to dryness stress make it challenging to develop accurate implementations (Green, 2024; Grossiord et al., 2020).





However, we note that our approach only captures the linear relationship of the local monthly response to SM changes and does not account for other factors potentially influencing the SM–induced effect on GPP (e.g., non–linear, non–local, or lagged effects). Nevertheless, the contribution analysis explains about 83 % (94 %) of global reduction in GPP due to SM changes for the LFMIP (GLACE–CMIP5) mean and is thus useful as an estimate of first–order processes influencing land carbon uptake (i.e., direct and indirect effects of SM).

## 4.3 Intermodel differences are dominated by the sensitivity of land carbon uptake to SM

The factorial ANOVA suggests that the sensitivity of GPP to direct and indirect SM effects is the main source of intermodel differences for LFMIP, especially in regions with agreement on SM drying (i.e., large parts of the Amazon, Central North America, and Central Europe). Differences in sensitivity to the direct SM effect contribute more to intermodel differences in the tropics, whereas that to indirect SM effects contributes strongly at mid to high latitudes.

The dominant role of sensitivity may partly result from differences in the implementation of water–stress related processes across LFMIP models (as mentioned in Sect. 4.2). However, the reasons underlying the intermodel differences are likely complex. In regions showing high agreement on SM drying across LFMIP for the future period, apparent divergence in the projected SM regimes (i.e., moisture–limited versus energy–limited regimes) may be a prominent cause of intermodel difference in the sensitivity of GPP to SM. It has been shown that water–carbon coupling is particularly strong in moisture–limited ecosystems, leading to an increased sensitivity of GPP to changes in SM (Gentine et al., 2019; Seneviratne et al., 2010). Hsu and Dirmeyer (2023) found substantial uncertainty in the simulated SM regimes of CMIP6 models. These uncertainties likely propagate to uncertainties in the sensitivity response of GPP.

In addition, differences in the representation of vegetation cover has important implications for carbon uptake. CMIP6 models exhibit substantial disagreement in vegetation cover, the simulation of disturbances and ultimately the handling of water stress (Song et al., 2021). This is further amplified by inconsistencies in the translation of land–use forcing data and the lack of explicit representation of forest management in ESMs (Egerer et al., 2025). Although the ability to represent vegetation structure has improved since CMIP5, limitations remain in the representation of disturbances, nutrient constraints, and forest demographics (Egerer et al., 2025; Gier et al., 2024). Thus, vegetation cover and the degree to which ecosystems are or will become moisture–limited could be a critical source of uncertainty in projections of SM–induced changes in land carbon uptake.

GLACE–like experiments offer a unique opportunity to isolate and assess the impact of SM changes on the evolution of the land carbon sink. Such information cannot be derived from observations because SM changes co–vary with other factors influencing the land carbon sink, thus isolating SM effects would require large–scale controls to fully capture SM–atmosphere interactions. Furthermore, benchmarking simulated SM and the resulting impact on the land carbon sink against past observations says little about satisfactory model performance under climate change, particularly since the effects strongly depend on the forcing scenario and human perturbations of the water and carbon cycles (Gier et al., 2024; Zaitchik et al., 2023).

However, we note that the validity of our results is limited by the small number of ESMs participating in LFMIP (and GLACE–CMIP5). Conducting the model experiment with a larger set of models would enhance the robustness and validity of the results and provide further insights into the effects of SM on the land carbon sink and associated uncertainties. We therefore



recommend continuing GLACE–like experiments in future CMIP generations with more models to generate a larger ensemble. Furthermore, we propose that a useful extension of the experimental setup could include an experiment with identically prescribed SM trend and variability (e.g., based on the CMIP MMM) to explicitly disentangle the contributions of SM effects
and model sensitivity of GPP to SM.

## 5   Conclusions

This study assesses the SM–induced impact on the land carbon sink using dedicated experiments that enable the isolation of SM effects from other drivers. We find that projected SM changes negatively impact global NBP in both CMIP5 and CMIP6 models, leading to a substantial reduction of the land carbon sink capacity by the end of the century. For both model generations,
SM variability dominates the negative impact, underscoring the key role of SM extremes in mediating the land carbon sink. The stronger impact in GLACE–CMIP5 is largely driven by a subset of models projecting severe SM drying.

Our analysis shows that indirect SM effects via T and VPD dominate global GPP losses, suggesting that SM–atmosphere coupling dominates the SM–induced reduction in global land carbon uptake. However, the contribution of direct and indirect effects varies strongly across models and regions. LFMIP models generally fail to capture the importance of the direct
SM effect in moisture–limited regions, which might be explained by their representation of water–stress related processes in ESMs. Differences in stomatal conductance schemes and the lack of hydraulic constraints could contribute to differences in the models' response of GPP to SM. This strong divergence in sensitivity across LFMIP models might be further amplified by uncertainties in projected SM regimes and vegetation characteristics, as well as misrepresentation of disturbance processes. Given that state–of–the–art ESMs lack accurate representation of processes from plant to ecosystem level, the true severity of
SM extremes on the land carbon sink is likely underestimated in LFMIP.

Our results highlight that SM–atmosphere coupling is a key constraint for the future land carbon sink. Multi–model intercomparison projects such as GLACE–CMIP5 and LFMIP (LS3MIP) constitute valuable tools to assess SM–induced changes in carbon uptake and related intermodel differences, by isolating the impact of SM from other drivers. We therefore urge to introduce again an experiment based on the GLACE–CMIP5/LFMIP set–up in the coming CMIP generations. Expanding the
number of participating models would further improve confidence in projections of SM–induced changes in the land carbon sink and help to further assess and refine underlying processes regulating water–carbon coupling.

*Data availability.* CMIP6 (including LFMIP) and CMIP5 data used in this study are publicly available from the Earth System Grid Federation (ESGF) at https://esgf-node.llnl.gov/search/. GLACE–CMIP5 data are hosted by ETH Zürich and can be obtained from S.I.S. (sonia.seneviratne@ethz.ch) for all participating models (see https://iac.ethz.ch/group/land-climate-dynamics/research/glace-cmip.html) or
from the individual climate modelling groups for individual models.



*Author contributions.* L.L. conceived the original idea. L.M.G., L.L., P.S., and S.I.S designed the study. L.M.G. carried out analyses under supervision of P.S., L.L., and S.I.S. L.M.G. wrote the first draft with contributions from P.S., and discussions with L.L. and S.I.S. All authors reviewed the manuscript and contributed to its final form.

*Competing interests.* The authors declare no competing interests.

*Acknowledgements.* We acknowledge the World Climate Research Programme and its Working Group on Coupled Modelling for coordinating and promoting the Coupled Model Intercomparison Project (CMIP), and the climate modelling groups for their contributions. We specifically thank all contributors to the LFMIP and GLACE–CMIP5 experiments.

This work received funding from the EU Project RESCUE (Response of the Earth System to overshoot, Climate neUtrality and negative Emissions) under the European Union's Horizon Europe research and innovation programme, grant agreement no. 101056939.

We also thank Mathias Hauser for downloading and preprocessing the data used for this analysis and for his technical support, and Martin Hirschi for discussions about the analysis.



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
