# Peer review of "Soil moisture—induced changes in land carbon sink projections in CMIP6"

_EGUsphere, 2025_

## Author Comment (AC2)

**Soil moisture-induced changes in land carbon sink projections in CMIP6,**
https://doi.org/10.5194/egusphere-2025-4215

**Response to Anonymous Referee #2**

*In this study, the authors compared the effects of soil moisture on GPP across CMIP5 and CMIP6 models. They found that SM variability is the main driver of land carbon reductions, and SM-atmosphere coupling is a critical factor for predicting future land carbon uptake. This research is complex and interesting, the manuscript is well-written, and suitable for the journal. The authors have put in a lot of work to complete the manuscript. However, there are some major concerns. Since this paper conducted a similar analysis to Green et al. (2019) and the main findings are similar, it is unclear what the innovative part of this study is. Secondly, the analysis is comprehensive and thorough, but missing uncertainty analysis, especially for the spatial maps (only shows the mean, without a uncertainties map). Thirdly, the method section needs significant improvement, especially regarding how the selected model simulates soil moisture, GPP, and how temperature affects GPP. Moreover, the authors did not explain whether the soil biogeochemical component of the model is included, as it is also an important part of carbon cycling. Did these models have the soil biogeochemical part? How will soil moisture affect microbial activity and Nitrogen availability? There is also a lack of a mechanism explanation for findings, please try to explain the mechanisms. Lastly, there are so many acronyms in the manuscript, which makes it very difficult to follow. Suggest making a list/table of the different acronyms. As a result, I would like to suggest a major revision.*

Thank you for the overall positive evaluation of our work. Your suggestions are very helpful to include additional uncertainty information, clarify the process representation in the considered LFMIP/CMIP6 models, and improve overall readability and organisation. Detailed responses are provided with the specific comments below.

On a general level, we would like to address the concern regarding the innovative part of this study compared with Green et al. (2019). In the first part of our manuscript (section 3.1), we analyse the results for the latest version of the SM experiment, LFMIP (based on CMIP6 models), and compare it with those for the previous version, GLACE-CMIP5 (based on CMIP5 models). To our knowledge, this is the first analysis of SM-induced impacts (trend and variability) on the land carbon sink using LFMIP. Furthermore, the comparison between CMIP5 and CMIP6 model generations has not been done before. The results are surprising as they demonstrate great differences in the magnitude of SM-induced impacts on the land carbon sink. This led us to the second part (section 3.2) where we investigate the origins of SM-induced changes in land carbon uptake (GPP), and whether intermodel differences are driven by differences in the SM evolution or the sensitivity of GPP to SM. This was not included in the study of Green est al. (2019).

To clarify these contributions, we will formulate research question at the end of the introduction:

1. Is the SM-induced impact on the land carbon sink (NBP) according to LFMIP (CMIP6) different from that according to GLACE-CMIP5 (section 3.1)?
2. Within LFMIP (CMIP6), what are the origins of (i) SM-induced changes in land carbon uptake (GPP) (section 3.2.1) and (ii) related intermodel differences (section 3.2.2)?

*Make a clear table showing the relationship between LFMIP and GLACE-CMIP5, as many acronyms for models or projects are making the manuscript very hard to read.*

We appreciate this suggestion. A table providing an overview over the experiments and included models is a very good idea. We will include information on experiment name (GLACE-CMIP5/LFMIP), CMIP generation (CMIP5/CMIP6), and available models (listing the four models available for GLACE-CMIP5 and LFMIP (CMIP6), respectively). Such a table will also give an overview of most of the acronyms used throughout the paper. This might make a dedicate acronym list obsolete, but we will consider that option too.

*Line 30: The concept of NBP is a little unclear, please elaborate more of this concept and what is the difference between NBP and NEP?*

The difference between NBP and NEP is that NBP also includes carbon loss due to disturbances (i.e., NBP = NEP - disturbances). Keenan and Williams (2018), which we refer to where NBP is introduced, provide detailed definitions for carbon sink related terms such as NEP and NBP. Introducing NEP in our paper would perhaps exceed the scope, as we analyse either NBP or GPP (but never NEP). Nevertheless, we understand that the current phrasing of the paragraph is confusing and we will improve this to make the concept of NBP clearer (i.e., NBP = GPP – respiration – disturbances).

*Line 35: temperature-induced increases in respiration, plants or soil microbials?*

In this line, "temperature-induced increases in respiration" refers to both, plant respiration and soil microbial respiration. We will rephrase the sentence to: "temperature-induced increases in respiration by plants and soil microbials, [...]". How SM affects respiration (plant and soil microbial respiration) will also be mentioned in this context.

*Line 50: Does this research include the below-ground soil-biogeochemistry or only the plant-vegetation part?*

All models participating in LFMIP (CMIP6) and GLACE-CMIP5 are fully coupled Earth System Models (ESMs) which all include both, below-ground soil-biogeochemistry and above-ground vegetation. We will mention this in the introduction.

*Line 57, this part is unclear, by further reducing SM, through what feedback loop? Higher transpiration or higher evaporation? Higher VPD may also cause a close of stoma to limit water loss, so it still decreases transpiration which may not reduce SM.*

We apologize for the confusion, this paragraph is poorly phrased. We will rephrase it to clarify the concept of indirect SM effects via temperature, VPD, and radiation on land carbon uptake.

*Suggest moving table S1 to the main text and making the ESM the first row instead of using the modelling group as the first row. What is the land model resolution? How many soil layers does each model have?*

Table S1 can be moved to the main text. The land model resolution will be added to the table. Information on soil layers can also be added. However, we highlight that we use standardized SM values (i.e., z-score) in our analysis and the focus lies on the effect of SM changes on the land carbon sink and not on the SM changes themselves or soil processes. Therefore, including the number of soil layers might not be of great importance for this analysis.

*Line 100, do not understand this paragraph, what is the purpose of selecting one ensemble member? Did the authors already select four models in the previous description?*

For the experiments GLACE-CMIP5 and LFMIP (CMIP6), we do not select models. Only four modeling groups participated with their respective model and ran one simulation (i.e., member) per experiment. Given the small number of available models, we use all four available models and evaluate their characteristics (e.g., SM, NBP) with respect to the CMIP ensemble consisting of 9-10 models. For this comparison we use the control run, which is identical to future projections simulated by all CMIP models (in contrast to the experiments with SM prescription).

Line 100 refers to the full ensemble of CMIP models, where multiple members (i.e., model runs) are available per model. We check if the available four models for the LFMIP/GLACE experiments (with only one member available) are representative for the larger CMIP5/CMIP6 ensembles using one member per model for comparability.

We will improve the text to clarify this and integrate the names of CMIP models used for comparison in the new table suggested under comment 1.

*Line 105, how is the NBP calculated?*

The models output NBP (i.e. GPP – respiration – disturbances). This will be better explained in introduction.

*Figure 1, there is a mismatch of caption descriptions and figure panels, please check and revise. The label of y-axis is a little confusing.*

Thank you for pointing this out. The caption of panel (a) and (b) were indeed swapped, this will be corrected. To avoid confusion on the y-axis label, we will make clear in the description (caption and title) of panel c) that it displays a 30-year mean. Consequently, all data is shown in the same unit (Pg C yr$^{-1}$).

*Line 181, "reduces the global land carbon sink to half", this sentence is confusing and difficult to understand.*

We understand that this is confusing, and we will try too rephrase it. Essentially it means that the negative impact of SM reduces the global land carbon sink to half of what it would be without the impact of SM (or in other words: the land carbon sink would be twice as strong without the negative impact of SM).

*Figure 1, why do you choose 20 20-year centred rolling mean and your analysis period is 29 years*

For the time series in panel a) and b) of Figure 1 we chose a 20-year rolling mean to remove the interannual variability and focus on the long-term trend without losing the features of the time series (which would happen when smoothing over longer periods). For the analysis of the impact of SM on NBP by the end of the century (panel c) we chose a 30-year period for averaging. A 30-year period is usually used to define "climate normals" because it is long enough to capture natural variability without being dominated by short-term weather fluctuations, and short enough for updating trends. In summary, 20 years are chosen for calculating trends and 30 years for calculating end of the century averages.

*Figure S4 is never mentioned or explained in the main text.\*

Thank you for pointing this out. Figure S4 will be mentioned in the results section where it is appropriate.

*Lines 205 – 210, the spatial analysis shows that the large SM differences between CMIP5 and LFMIP for the IPSL and GFDL models. Can you explain what is causing the differences. Moreover, if the ensemble of many models means kind of similar, this indicates the IPSL and GFDL may not be typical representative models, so why do you select these two models?*

Thank you for this question. As mentioned under an earlier comment, we do not select models but use the four available models for GLACE-CMIP5 and LFMIP, respectively, and evaluate their characteristics (e.g., SM, NBP) with respect to the CMIP ensemble consisting of 9-10 models. This will be clarified in the methods.

When comparing SM of IPSL and GFDL against a larger set of CMIP5 models we find that they are not representative of the CMIP5 multi-model mean. However, this does not mean that these models are no plausible and should be removed. The purpose of the comparison is to put the

available models into context and provide an overview on where they lie within the ensemble (e.g., are they among drying or wetting models, what do their spatial patterns of drying/wetting look like). This is important information to consider when interpreting the results. Furthermore, by comparing SM within and across model generations, we acknowledge that models and generations differ. We will also clarify these aspects in the text.

Why the models project different SM evolution is outside the scope of our work. However, other papers investigate that in more detail (e.g. Berg et al. (2016) - Divergent surface and total soil moisture projections under global warming,  Cook et al. (2020) - Twenty-First Century Drought Projections in the CMIP6 Forcing Scenarios, or Wu et al. (2024) - Hydrological Projections under CMIP5 and CMIP6). We will refer the reader to these references.

*Figure 2 and 3 may need to perform some uncertainty analysis of the spatial patterns.*

Thank you for this comment. This suggestion will be implemented. For the model-specific plots we will add information on trends by adding stippling on where trends over time are (in)significant. For model mean plots, hatching for sign disagreement will be added.

*Lines 212, be specific, how many available models? Please also describe this more clear in the method section.*

We apologize for the lack of clarity. We will add "of the four available models" for the GLACE–CMIP5 and LFMIP and also add numbers for the CMIP5/6 ensembles (ten and nine, respectively) in this paragraph. The numbers of available models are mentioned in the methods section already, but your suggestion to include a table on the experiments (see comment 1) will be helpful to provide a better overview.

*Figure 5, please include the meaning of $GPP_R$ in the caption too. Please also explain in the text why the indirect effect R from SM leads to an increase in GPP. Can the selected model capture the optimum temperature effect for plant GPP.*

$GPP_R$ is mentioned in the caption already, but we apologize for using the acronyms to explain it. The caption will be edited such that the meaning of each term is clear. Furthermore, we will explain in the text how indirect SM-effects via radiation can increase GPP. The models capture the effect of temperature on GPP, since photosynthesis is implemented as a function of radiation, temperature, water (and nutrient) availability and CO2. As suggested, we will add a brief explanation on how the variables of interest (e.g., soil moisture, GPP, NBP) are simulated when introducing the models in the methods section. In table S1 we list references for each model, which provide further information on how the models are built, with detailed explanations of how specific processes are implemented conceptually and mathematically per model.

*Lines 315-320, please be specific, "some land surface models" which models? Maybe in the method section describe a little how the GPP and soil moisture are modelled in the four selected models.*

We agree that this formulation sounds vague and needs to be rephrased. It does not refer to specific models used in this analysis, but to state-of-the-art land surface models in general (as shown by Franks et al., 2018), which are also used in LFMIP models. The aim was to point out that the process formulation of stomatal condutance in state-of-the-art ESMs is often linked to VPD. This could be one explanation why indirect effects are found to be a major driver of SM-induced changes of GPP in our analysis. We will edit this part for clarity. As suggested, we will also add a brief explanation on how the variables of interest are simulated when introducing the models in the method section.

*Line 369: How to quantify the SM variability quantitatively?*

The existing GLACE-CMIP5/LFMIP experimental setup allows isolating the model-specific effect of SM trend and variability on the land carbon sink (from any other future changes in the land carbon sink). However, it does not allow separating the effects of differences in SM evolution and sensitivity to SM across models. Therefore, in line 369, we suggest additional experiments with identical SM evolution, e.g., using the CMIP MMM of soil moisture (corresponding to SM in the CTL experiment). Based on that, an identical SM trend (similar to rmLC) and an identical prescribed mean seasonal cycle of SM (similar to pdLC) could be implemented. Such experiments enable to isolating the effect of identical SM trend and variability on the land carbon sink which would help to assess intermodel differences in in SM-induced changes of NBP in further detail. We understand that the current phrasing is too short and we will rephrase this paragraph to better explain this idea.